# In vivo dual RNA-seq reveals that neutrophil recruitment underlies differential tissue tropism of *Streptococcus pneumoniae*

Vikrant Minhas[1,4], Rieza Aprianto [2,4], Lauren J. McAllister[1], Hui Wang[1], Shannon C. David[1], Kimberley T. McLean[1], Iain Comerford[3], Shaun R. McColl[3], James C. Paton [1,5✉], Jan-Willem Veening [2,5] & Claudia Trappetti[1,5]

*Streptococcus pneumoniae* is a genetically diverse human-adapted pathogen commonly carried asymptomatically in the nasopharynx. We have recently shown that a single nucleotide polymorphism (SNP) in the raffinose pathway regulatory gene *rafR* accounts for a difference in the capacity of clonally-related strains to cause localised versus systemic infection. Using dual RNA-seq, we show that this SNP affects expression of bacterial genes encoding multiple sugar transporters, and fine-tunes carbohydrate metabolism, along with extensive rewiring of host transcriptional responses to infection, particularly expression of genes encoding cytokine and chemokine ligands and receptors. The data predict a crucial role for differential neutrophil recruitment (confirmed by in vivo neutrophil depletion and IL-17 neutralization) indicating that early detection of bacteria by the host in the lung environment is crucial for effective clearance. Thus, dual RNA-seq provides a powerful tool for understanding complex host-pathogen interactions and reveals how a single bacterial SNP can drive differential disease outcomes.

[1] Research Centre for Infectious Diseases, Department of Molecular and Biomedical Science, University of Adelaide, Adelaide 5005, Australia. [2] Department of Fundamental Microbiology, Faculty of Biology and Medicine, University of Lausanne, 1015 Lausanne, Switzerland. [3] Department of Molecular and Biomedical Science, University of Adelaide, Adelaide 5005, Australia. [4]These authors contributed equally: Vikrant Minhas, Rieza Aprianto. [5]These authors jointly supervised this work: James C. Paton, Jan-Willem Veening, Claudia Trappetti. ✉email: james.paton@adelaide.edu.au

Streptococcus pneumoniae is a major human pathogen responsible for massive global morbidity and mortality. Despite this, the pneumococcus makes up the part of the commensal human nasopharyngeal flora, colonizing up to 65% of individuals[1,2]. S. pneumoniae can invade from this reservoir to cause disease, for example, by aspiration into the lungs to cause pneumonia, by invasion of the blood (bacteremia) or central nervous system (meningitis), or by ascension of the eustachian tube to cause otitis media (OM)[1,2]. S. pneumoniae is an extremely heterogeneous species, comprising at least 98 capsular serotypes and over 12,000 clonal lineages (sequence types; ST) recognizable by multi-locus sequence typing[3,4]. Unsurprisingly, S. pneumoniae strains differ markedly in their capacity to progress from carriage to disease and/or the nature of the disease that they cause[1,2].

We have previously reported marked differences in virulence in a murine intranasal (IN) challenge model between S. pneumoniae strains belonging to the same serotype and ST, which correlated with clinical isolation site in humans (ear versus blood). In serotype 3 ST180, ST232, and ST233, and in serotype 14 ST15, human ear isolates had greater capacity to cause OM in mice relative to their respective serotype-/ST-matched blood isolates, while blood isolates preferentially caused pneumonia or sepsis in mice, suggesting stable niche adaptation within a clonal lineage[5,6]. Recently we have shown that although the genomes of serotype-/ST-matched blood and ear isolates differ by several single nucleotide polymorphisms (SNPs), the distinct virulence phenotypes correlated with single SNPs in genes encoding uptake and utilization of the sugar raffinose[7]. In serotype 14 ST15, the SNP was in the regulatory gene rafR, while in serotype 3 ST180, the SNP was in rafK, which encodes the raffinose ABC transporter ATPase[7]. Both SNPs result in non-conservative amino acid changes in functionally critical domains of the respective gene product (D249G for RafR; I227T for RafK). Moreover, in both serotypes/lineages, ear isolates had in vitro growth defects in a chemically-defined medium with raffinose as sole carbon source, correlating with defective transcription of raffinose pathway operons. Remarkably, in serotype 14 ST15, exchanging the rafR alleles between blood and ear isolates reversed both the in vitro and in vivo phenotypes[7]. Thus, the D249G SNP in rafR appears to be the determinant of differential virulence phenotype between the blood and ear isolates of this lineage, which may reflect differential engagement of innate host defenses and/or differential bacterial nutritional fitness in distinct host niches (Fig. 1).

Dual RNA-seq applies deep sequencing to simultaneously quantify genome-wide transcriptional responses of host and pathogen[8,9]. This approach offers higher efficiency and more restricted technical bias compared to conventional approaches, such as assaying single species or array-based methods. Here, we have used dual RNA-seq analysis to examine host-pathogen transcriptional cross-talk in the blood and ear isolates and rafR-swapped derivatives thereof, during the early stages of infection. Our data strongly suggest that the rafR SNP interacts with the pneumococcal genetic background in the different clinical isolates, which in turn, induces variegated transcriptional responses in the pathogen; this response, in turn, initiates a diverging host response that determines the outcome of infection.

## Results

**Comparative host/pathogen transcriptomics.** Our previous studies have shown that at 6 h after IN challenge with serotype 14 ST15 S. pneumoniae, numbers of blood and ear isolates (strains 4559-Blood and 9–47-Ear, respectively) in murine lungs are similar ($10^6$–$10^7$ CFU per lung). However, by 24 h, the ear isolate had been cleared from the lungs, instead spreading to the ear and brain. In contrast, the blood isolate persisted in the lungs at 24 h, but did not spread to the ear or brain[6]. Thus, 6 h post infection is a critical decision point in the pathogenic process, and the similar bacterial loads in the lung at this time enables examination of host/pathogen transcriptional cross-talk without the complication of bacterial dose effects. Accordingly, groups of 12 mice were anaesthetized and challenged IN with $10^8$ CFU of 4559-Blood, 9–47-Ear or their respective rafR-swapped mutants; at 6 h, mice were euthanized and total RNA was extracted from perfused lungs and purified. RNA from lungs of four mice was pooled for dual RNA-seq analysis in triplicate (see "Methods" section).

Within the sequencing libraries, an overwhelming majority of reads originate from the host genome (average 99.5%; range 99.1–99.7%), which translates into an average depth of 1.3 times (range 0.8–1.8 times). Conversely, 0.52% of the total reads originated from the pathogen genome (0.33–0.93%) (Supplementary Table 1). Of these pneumococcal reads, 64.5% (61.4–67.3%) mapped onto rRNA genes and 35.5% (32.7–38.6%) mapped onto non-rRNA genes. Previous data indicate that non-depleted libraries only contain 5% non-ribosomal RNA reads; thus, this treatment enriched the non-ribosomal RNAs sevenfold. Non-ribosomal read depth was 2.7 times (1.4–4.6 times) for the pathogen genome. Further downstream analysis, including

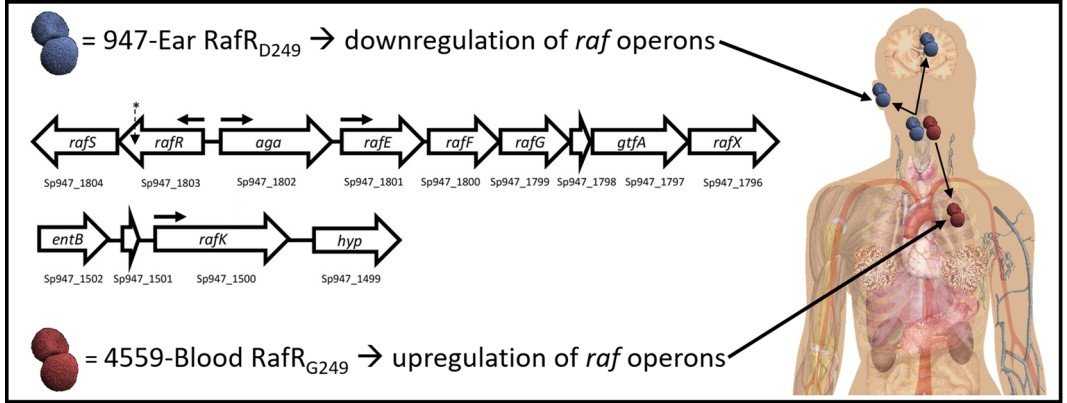

**Fig. 1 Impact of rafR SNP.** A SNP in rafR between the serotype 14 sequence type 15 clonal isolates 4559-Blood and 947-Ear leads to a non-conservative G249D amino acid substitution in the raffinose pathway regulator RafR. RafR$_{G249}$ results in upregulation of raf operons (horizontal arrows denote transcriptional start sites) in 4559-Blood relative to 947-Ear, favouring persistence in the lung after intranasal challenge. Lower raf pathway expression mediated by RafR$_{D249}$ facilitates clearance of 947-Ear from the lung, but promotes spread to and/or persistence in the ear and brain. The location of the SNP in rafR is indicated by an asterisk[7].

differential gene expression, excluded ribosomal reads from the pathogen library. Supplementary Data 1–6 list pneumococcal genes that are significantly differentially expressed (fold change (FC) > 2, $p < 0.05$) for each of the six pairwise comparisons between the four strains. Supplementary Data 7–12 list murine genes that are significantly differentially expressed (FC > 1.5, $p < 0.05$) for the same pairwise comparisons. The lower FC cut-off for murine genes was chosen to strike a balance between simplicity of analysis and sensitivity. Heat maps displaying the top 50 murine and top 50 pneumococcal genes (Fig. 2) showcase the breadth of transcriptomic rewiring due to the *rafR* SNP.

**RafR fine tunes carbohydrate metabolism during infection**. In order to directly compare pathogen transcriptional responses in murine lung, we listed homologous genes between the wild type ear and blood isolates and used these to visualize the transcriptional response in a principal component analysis (PCA) plot (Fig. 3a). Here, the pneumococcal transcriptional response of the ear isolate (strain 9–47-Ear, dark orange) to murine lung infection diverges considerably from the response of the blood isolate (strain 4559-Blood, dark purple). Specifically, 76 homologous genes are upregulated in the ear isolate, while 40 are upregulated in the blood isolate. Upregulated genes in strain 9–47-Ear include genes involved in carbohydrate metabolism, general stress response and nutrient transporters, while upregulated genes in the blood isolate include genes encoding small molecule permeases and nisin biosynthesis orthologous proteins.

Furthermore, replacing *rafR* of 9–47-Ear with the allele from the 4559-Blood (designated strain 9–47M) dissociates its transcriptional response considerably from its parental strain (Fig. 3a; 9–47-Ear, dark orange to 9–47M, light orange). These large scale transcriptomic differences between 9–47-Ear and 9–47M may explain the impact of the *rafR* SNP on in vivo tissue tropism. Specifically, 87 genes are upregulated in the wild type strain (9–47-Ear) while 36 genes are upregulated in the otherwise isogenic *rafR* swap strain (9–47M, Fig. 3b), with differentially expressed genes spread across the pneumococcal genome. Presence of the blood isolate *rafR* allele in 9–47-Ear activates the expression of major genes pertaining to carbohydrate metabolism, including *adhA* (alcohol dehydrogenase) and *spxB* (pyruvate oxidase), as well as genes encoding permeases, including *glnH6P6* (transporting arginine, cysteine) and *ycjOP-yesO* (transporting multiple sugars). Also, a subset of genes with function in carbohydrate metabolism are repressed in the *rafR*-swap strain, such as glycogen synthesis (*glgACD*), sucrose metabolism (*scrB*) and ribulose metabolism (*ulaDEF*). Expression of seven genes encoding ATPase subunits (*ntpABCDEGK*) and genes for iron (*piuB*) and sugar (*scrA, satABC,* and *gadEW*) permeases are also repressed.

On the other hand, replacing the blood isolate *rafR* with the ear allele (strain 4559M) does not noticeably interrupt pneumococcal transcriptional response to murine lung (Fig. 3a; 4559-Blood, dark purple to 4559M, light purple), despite the different in vivo behaviors of 4559-Blood and 4559M. Essentially, the *rafR* swap activates only two genes: *yxlF*, encoding a putative subunit of an ABC transporter and *phoU1* encoding a phosphate transporter; and represses 35 genes, mostly contained in a single genomic island (Fig. 3c, upregulated in 4559-Blood). The genomic island consists of 28 consecutive genes encoding subunits of bacteriophage(s), interspaced by *dnaC*, encoding a DNA replication protein and *lytA*, encoding autolysin. The activation of bacteriophage-associated genes indicates that the original isolate (strain 4559-Blood) endures host-derived stress, unlike the *rafR* swap mutant (strain 4559M). Other genes repressed in 4559M include *adhAE* (alcohol dehydrogenases), *gtfA* (sucrose

phosphorylase) and *rafEG* (raffinose transporter). Conversely, 9–47M and 4559-Blood showcase similar in vivo behaviors yet display varied transcriptomes in murine lungs post-infection (Fig. 3a). Hence, transcriptional profiling was unable to predict tissue tropism in this case. Taken together, the single D249G SNP in *rafR* interferes with global gene expression within the already transcriptionally-distinct parental clinical isolates. This effect is more pronounced in 9–47-Ear than 4559-Blood.

Next, we performed quantified enrichment analyses on specific gene functions. Carbohydrate metabolism is enriched in the differentially expressed genes between the strains, particularly when comparing the ear isolate to its cognate *rafR* swap (Fig. 3d; *comparison A*, 9–47-Ear versus 9–47M, $p = 0.03$), comparing the two clinical isolates (*comparison B*, 9–47-Ear versus 4559-Blood, $p = 0.017$) and comparing the swap cognates (*comparison E*, 9–47M versus 4559M, $p = 0.041$). Another function, ABC transporters, is also enriched in the comparison within the ear isolates (Fig. 3e; *comparison A*, 9–47-Ear versus 9–47M, $p = 0.049$), between the original isolates (*comparison B*, 9–47-Ear versus 4559-Blood, $p = 0.014$), between ear isolate and *rafR* 746G in the blood isolate background (*comparison C*, 9–47-Ear versus 4559M, $p = 0.01$) and between the *rafR* cognates (*comparison E*, 9–47M versus 4559M, $p = 1.8 \times 10^{-4}$).

Additionally, since the pneumococcal genome has an exceptionally high number of sugar transporters[10], we quantified enrichment for this function (Fig. 3f). Sugar transporters are enriched in almost all comparisons (except between 4559-Blood and 4559M), highlighting the role of *rafR* in the widespread regulation of pneumococcal sugar importers. Specifically, ear and blood isolates behave differently in regard to sugar transporter expression (9–47-Ear versus 4559-Blood; Fig. 3f, *comparison B*). The ear isolate upregulates *scrA* (encoding a mannose and trehalose transporter) and *ulaA* (ascorbate transporter), while the blood isolate upregulates *ycjOP-yesO* (alternative sugar transporters), *rafE* (raffinose transporter) and *malFG* (maltose transporter). Furthermore, *rafR* swap in the ear isolate background (9–47-Ear versus 9–47M; Fig. 3f, *comparison A*) reduces the expression of *gadEW* (encoding sorbose and mannose transporter), *satABC* (arabinose and lactose transporter), *ulaAC* and *glpF* (glycerol transporter), while the swap activates the expression of *ycjOP-yesO* and *bguD* (encoding complex polysaccharide transporters). In contrast, *rafR* swap in the blood isolate background (4559-Blood versus 4559M; Fig. 3f, *comparison F*) downregulates the expression of *rafEG* and *malD* (maltose transporter). The enrichment analysis reveals that the D249G SNP in *rafR* directly and indirectly affects the expression of genes encoding sugar transporters, other (ABC) transporters and carbohydrate metabolism.

We also identified genes that were commonly up or down-regulated between the strains with a given virulence phenotype (Supplementary Table 2). *adhP* (Sp947_00279) was upregulated in both strains that persisted in lungs (9–47M and 4559-Blood), while eight genes from the genomic region Sp947_0842 to Sp947_0855, as well as Sp947_00631 and Sp947_02096, were upregulated in strains that were cleared from lungs by 24 h (9–47-Ear and 4559M). These genes include neuraminidase *nanB*, and alpha-glycerophosphate oxidase *glpO* (Supplementary Table 2). All other differentially expressed pneumococcal genes are listed in Supplementary Data 1–6.

**RafR-specific rewiring of host transcriptional responses**. The measured murine transcriptional response represents aggregate gene expression of all (host) cells present during pneumococcal lung infection. These include epithelial cells, endothelial cells of lung vasculature, smooth muscle cells, fibroblasts, activated, and

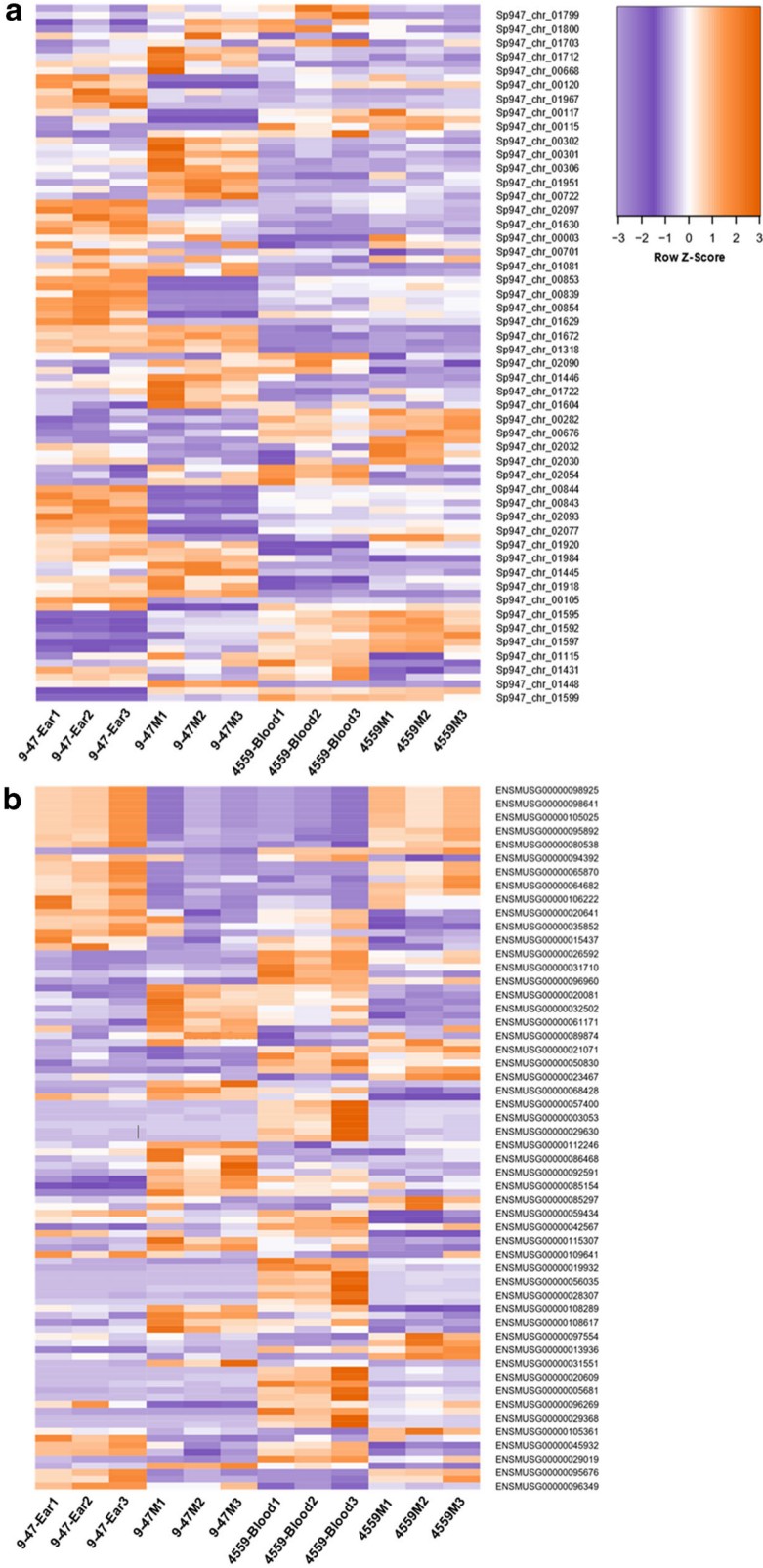

**Fig. 2 Heat maps displaying differentially expressed genes from the dual RNA-seq data in murine lungs 6 h after infection with either 9–47-Ear, 4559-Blood, 9–47M, or 4559M.** Purple indicates relatively low expression, while orange indicates relatively higher expression values. Top 50 differentially expressed pneumococcal (**a**) and murine (**b**) genes are shown. Fold change values of all statistically significantly expressed genes can be found in Supplementary Data 1–12.

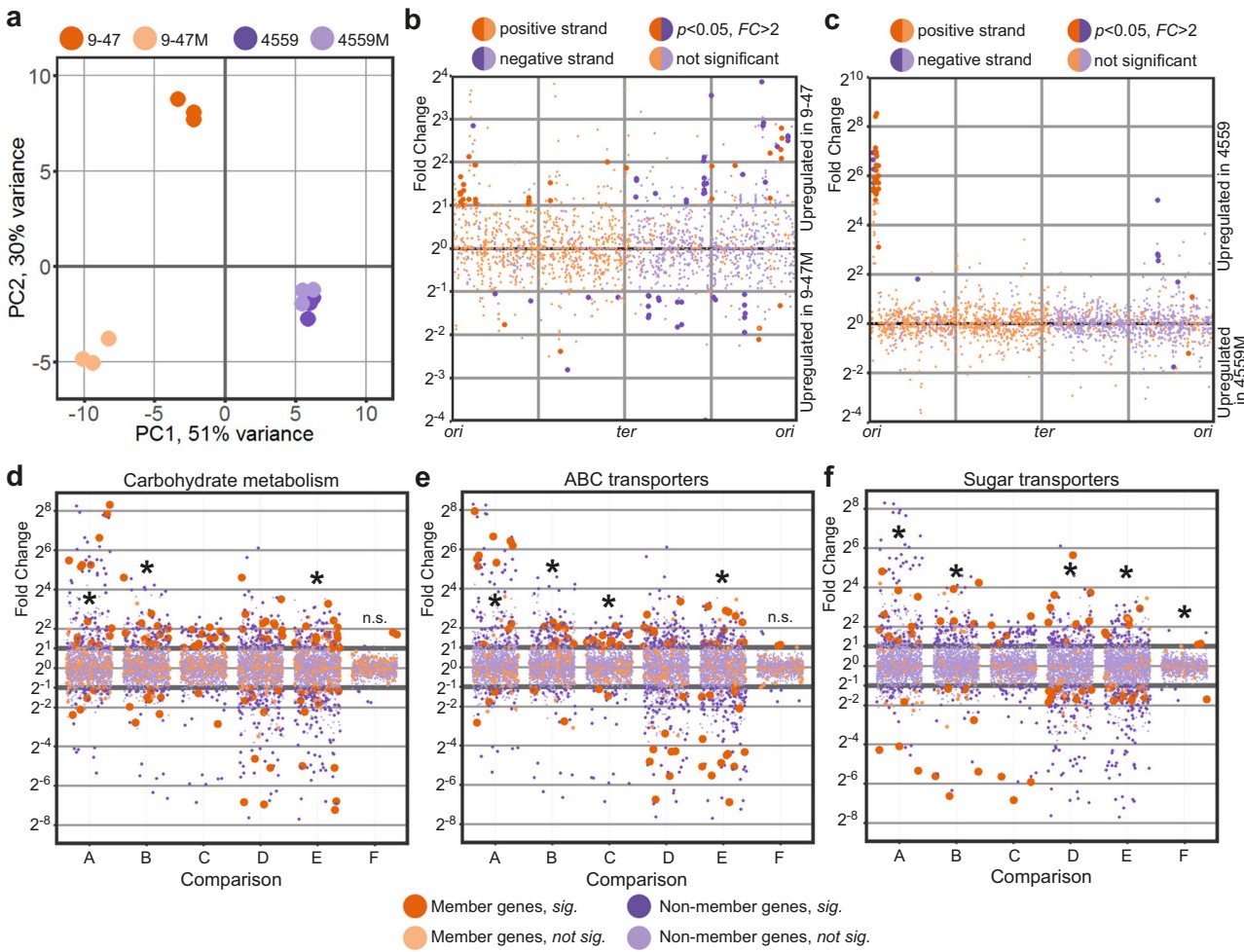

**Fig. 3 Pathogen transcriptional responses in murine lung. a** PCA plot showing divergence of transcriptional response to lung infection within the ear (9–47-Ear) and blood isolates (4559-Blood). *rafR* swap (9–47M) rewires pneumococcal transcriptional response only in the ear isolate background but not in the blood isolate. **b** Differential expression due to the *rafR* swap in the ear isolate background is spread throughout the pneumococcal genome. **c** Differential expression due to the *rafR* swap in the blood isolate background is limited to a genomic island. **d**–**f** Functional enrichment showed specific function being differentially expressed, including carbohydrate metabolism (**d**), ABC transporters (**e**), and sugar transporters (**f**). A: comparison of 9–47-Ear to 9–47M; B: 9–47-Ear to 4559-Blood; C: 9–47-Ear to 4559M; D: 9–47M to 4559-Blood; E: 9–47M to 4559M and F: 4559-Blood to 4559M. Asterisk (\*) denotes statistically significant functional enrichment for the indicated strain–strain comparison.

non-activated immune cells. The host transcriptional response was specific to the infecting pneumococcal strain (Fig. 4a). Specifically, there was a diverging host response to the ear isolate (9–47-Ear, dark orange) and blood isolate (4559-Blood, dark purple). Interestingly, *rafR* swap in blood isolate background (4559M, light purple) mimics the lung response to the wild type ear isolate (9–47-Ear, dark orange); these two strains harbor the D249 *rafR* allele. Surprisingly, the *rafR* swap in the 9–47-Ear background (9–47M, light orange) which harbors the G249 allele, does not drive the host response to mimic those of the wild type 4559-Blood strain (dark purple) that also has the G249 allele, but rather towards a new, third position of genome-wide expression.

Genome-wide plotting of murine transcriptional responses to *S. pneumoniae* strains 9–47-Ear and 4559-Blood shows extensive rewiring of gene expression across the murine chromosomes, consistent with their different tissue tropisms in vivo (Fig. 4b). Specifically, 433 murine genes are activated upon infection by 9–47-Ear (FC > 1.5, $p < 0.05$), while 787 genes are activated by infection with 4559-Blood (FC > 1.5, $p < 0.05$). Of the 9–47-Ear-upregulated murine genes, only 37% were protein-coding, with the majority encoding pseudogenes and small RNAs. Conversely, of the 787 4559-Blood-upregulated murine genes, 80% were

protein-coding, while the rest encoded small RNAs. The 4559-Blood-upregulated genes encode proteins involved in multiple pathways such as general metabolism, peroxisome proliferator-activated receptor (PPAR) signaling, steroid hormone biosynthesis and cAMP signaling. Although both strains belong to the same capsular serotype and ST[6], our data strongly suggest wildly diverging isolate-specific host responses during early infection, with the *rafR* SNP varying the host response considerably. Interestingly, the host transcriptional response to 4559M closely resembles that of 9–47-Ear (Fig. 4a), suggesting that the *rafR* SNP plays a crucial role in the distinct response to 9–47-Ear versus 4559-Blood. However, the host transcriptional response to 4559-Blood does not closely resemble the response to 9–47M, despite the similar in vivo tissue tropism these strains display. This suggests, unsurprisingly, that the host transcriptional response to pneumococcal infection does not perfectly correlate with in vivo outcomes.

In addition, *rafR* swap in the ear isolate background (9–47M) expressing the G249 *rafR* allele activates 271 murine genes (FC > 1.5, $p < 0.05$), while it represses 479 genes. The G249 *rafR*-activated genes include those involved in the Wnt signaling pathway (*Fzd2, Lgr6, Rspo1, Sost, Sox17, Wnt3a, Wnt7a*) and

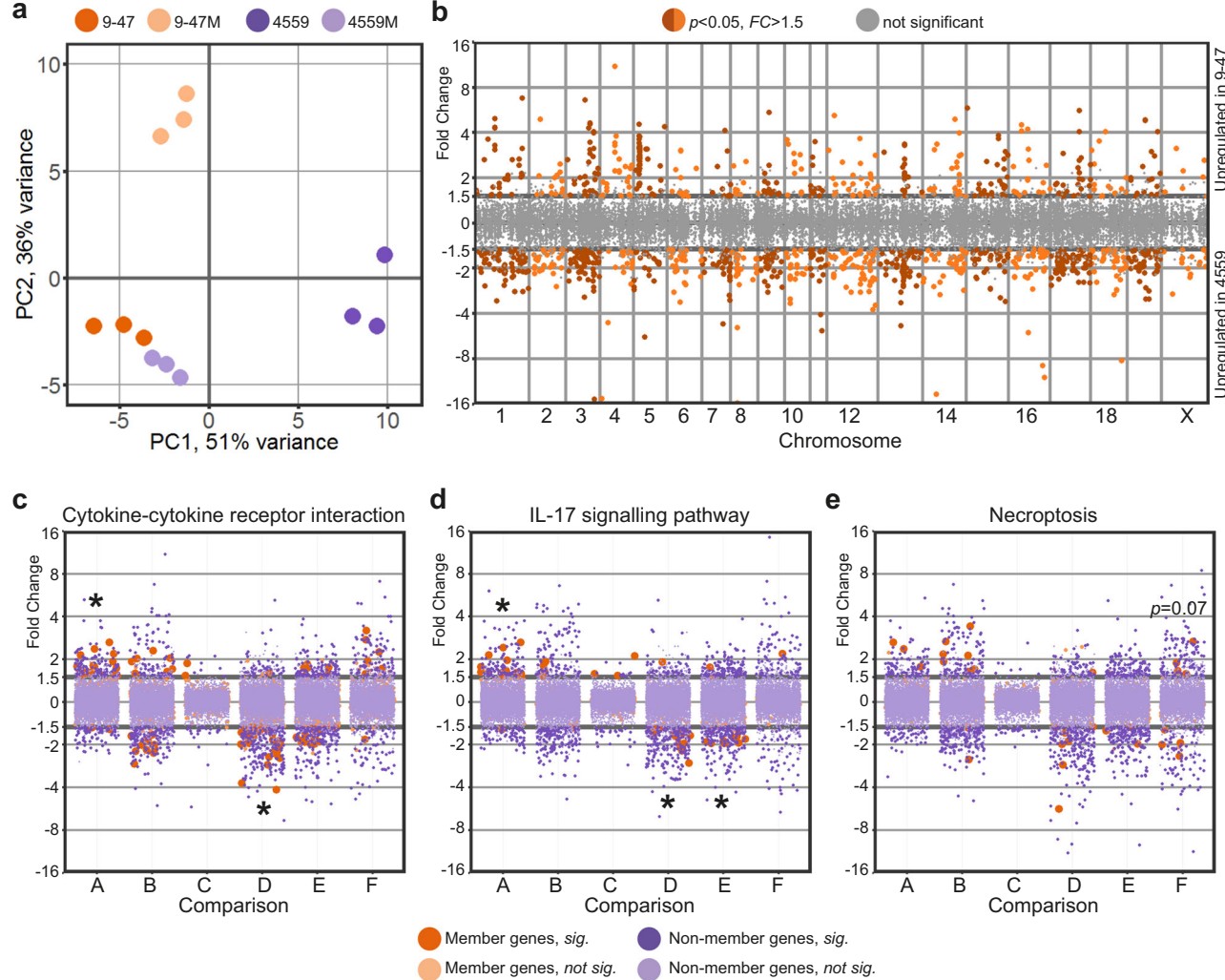

**Fig. 4 The SNP in pneumococcal *rafR* drives diverging host response. a** PCA plot illustrates murine lung response to the pneumococcal strains. Interestingly, host transcriptional response to *rafR* swap in blood isolate (4559M, light purple) is similar to the murine response to the original ear strain (9–47-Ear, dark orange). **b** Differential gene expression of transcriptional response to pneumococcal ear and blood isolates shows a widespread transcriptional rewiring. Specifically, 433 genes are activated in response to infection by ear isolate (9–47-Ear) while 787 genes are activated (FC > 1.5, p < 0.05) by blood isolate (4559-Blood). **c**–**e** Specific gene ontology terms are enriched in differentially expressed host genes in response to pneumococcal infection: cytokine–cytokine receptor interaction (**c**), interleukin-17 signaling pathway (**d**), and necroptosis (**e**). A: comparison between 9–47-Ear to 9–47M; B: 9–47-Ear to 4559-Blood; C: 9–47-Ear to 4559M; D: 9–47M to 4559-Blood; E: 9–47M to 4559M and F: 4559-Blood to 4559M. Asterisk (*) denotes statistically significant functional enrichment for the indicated strain–strain comparison.

general calcium signaling pathway (*Adra1a, Adra1b, Adrb3, Cckar, Grin2c, P2rx6, Tacr1, Tacr2*). Conversely, 52% of the G249 *rafR*-repressed genes in lungs infected with 9–47M encode RNA features and 18 chemokines, chemokine ligands, interferons and interleukins. On the other hand, *rafR* swap in the blood isolate background (4559M) expressing the D249 *rafR* allele activates 328 murine genes (FC > 1.5, p < 0.05), and represses 472 genes. Seventy-three percent of the D249 *rafR*-activated murine genes encode RNA features and 33 encode histone proteins. The activation of these histone proteins suggests a massive reorganization of gene regulation with numerous potential downstream impacts. In contrast, D249 *rafR*-repressed genes include genes encoding calmodulins (*Calm4, Calm13,* and *Camk2a*) and phospholipases A2 (*Pla2g4b, Pla2g4d,* and *Pla2g4f*).

Moreover, there are only 132 differentially expressed host genes in response to wild type 9–47-Ear compared to the response to strain 4559M (both having the D249 *rafR* allele), with 38 genes upregulated in strain 9–47-Ear and 94 in strain 4559M. Intriguingly, the D249 *rafR* allele (strains 9–47-Ear and 4559M) is

associated with a upregulation of RNA features, including antisense, intronic, long intergenic non-coding RNAs (lincRNAs) and micro RNAs (miRNAs). The resulting abundance of RNA species in murine cells upon pneumococcal infection has the potential for even more widespread transcriptional rewiring and fine-tuning of gene products later in the infection.

A quantified functional enrichment showed that certain gene functions are enriched in the murine response to pneumococcal strains. In particular, cytokine-cytokine receptor interaction is enriched in differentially expressed host genes because of *rafR* swap in the ear isolate background (Fig. 4c, *comparison A*, 9–47-Ear versus 9–47M, $p = 9.5 \times 10^{-4}$). Concurrently, the function is enriched in differentially expressed genes between mice infected with 9–47M and those infected with 4559-Blood (*comparison D*, $p = 1.8 \times 10^{-4}$). Since both strains harbor the G249 *rafR* allele, the differentially expressed genes encoding cytokines and cytokine receptors are most likely attributable to unrelated genetic differences between the clinical isolates. Interestingly, this function is not enriched in differentially expressed genes

between the *rafR* swap in the blood isolate background (4559M) and the wild type 9–47-Ear (*comparison C*), both of which have the D249 *rafR* allele. Genes encoding chemokine ligands (*Cxcl2, Cxcl3, Cxcl10,* and *Ccl20*), interleukin 17F (*Il17f*), interferon beta (*Ifnb1*) and a receptor of TNF (*Tnfrsf18*) are the common differentially expressed genes in lungs of mice infected with 9–47-Ear, 9–47M, and 4559-Blood, with ascending expression from responses to 9–47M, 9–47-Ear and 4559-Blood. Other genes encoding chemokine ligands (*Ccl3, Ccl4, Ccl17, Ccl24, Cxcl5, Cxcl11,* and *Xcl1*), interleukins (*Il1rn* and *Il13ra2*) and interferon gamma (*Ifng*) are more highly expressed in the ear isolate-infected lung (9–47-Ear) than in lungs infected by the *rafR* swap ear isolate (9–47M). Finally, genes encoding chemokine receptors (*Ccr1* and *Ccr6*), interleukin receptors (*Il1r2, Il10ra, Il17a, Il18rap, Il20ra, Il20rb, Il22,* and *Il23r*), and interleukins (*Il1f5, Il1f6, Il1f8,* and *Il6*) are more highly expressed in lungs infected by 4559-Blood compared to the *rafR* swap in the ear isolate background (9–47M).

Interleukin 17, as part of the cytokine response, activates multitudes of downstream targets in defense against infectious agents[11], and thus plays a central role in host response to pneumococcal infection. Here, we observe the same pattern of diverging activation among murine responses to the pneumococcal strains (Fig. 4d), with IL-17-associated genes being enriched in differentially expressed genes among the host transcriptional response to the *rafR* swap in the ear isolate background (*comparison A*, 9–47-Ear versus 9–47M, $p = 9.5 \times 10^{-4}$). These genes are also enriched amongst the host response to pneumococcal strains with the G249 *rafR* allele (*comparison D*, 9–47M versus 4559-Blood, $p = 7.5 \times 10^{-4}$) and to the *rafR* swap cognates (*comparison E*, 9–47M versus 4559M, $p = 0.022$). However, there was no enrichment of IL-17 associated genes amongst the host response to pneumococcal strains with the D249 *rafR* allele (*comparison C*, 9–47-Ear versus 4559M). Common differentially expressed genes of this function include genes encoding interleukin 17F (*Il17f*) and chemokine ligands (*Cxcl2, Cxcl3* and *Ccl20*), with ascending expression level of response to 9–47M, 9–47-Ear, and 4559-Blood. Specifically, the products of these genes regulate the recruitment of neutrophils and activate immune responses to extracellular pathogens.

In addition to the above, necroptosis, a programmed cell death, exhibits a trend for its enrichment ($p = 0.07$) in differentially expressed murine genes because of the *rafR* swap in the blood isolate background (Fig. 4e, *comparison F*, 4559-Blood versus 4559M). Genes encoding histone cluster 2 (*Hist2h2ac, Hisr2-h2aa1, Hist2h2aa2,* and *H2afx*) are more highly expressed in murine lungs infected with 4559-Blood, while those encoding phospholipases A2 (*Pla2g4b, Pla2g4f,* and *Pla2g4d*) and a subunit of calcium/calmodulin-dependent protein kinase II (*Camk2a*) are more highly expressed in the transcriptional response to the *rafR* swap in the blood isolate background (4559M).

**Validation of host/pathogen transcriptomics**. To validate the findings from the dual RNA-seq, quantitative real time RT-PCR was performed on the same RNA samples from the lungs 6 h post-infection. 19 pneumococcal and 18 murine genes were chosen for this validation, with the primers used listed in Supplementary Table 3. A total of 76 pneumococcal and 72 murine gene $Log_2$ FC comparisons were performed, with a high degree of correlation observed for both pneumococcal ($R^2 > 0.73$, Pearson) and murine genes ($R^2 > 0.81$, Pearson) (Fig. 5a). To validate the reproducibility of the dual RNA-seq data, lung RNA samples 6 h post-infection from a different experiment were analysed with qRT-PCR. Here, 6 pneumococcal and 6 murine genes were chosen, and 24 pneumococcal and 24 murine gene $log_2$ FC

comparisons were performed. A high degree of correlation was also observed here, for both pneumococcal ($R^2 > 0.77$, Pearson) and murine genes ($R^2 > 0.73$, Pearson) (Fig. 5b).

**Immune cell subsets present in infected lung tissue**. The host RNA-seq data represent the pooled transcriptional responses of all cell types present in the lungs at the time of RNA extraction. Thus, some of the transcriptomic differences may be attributable to alterations in the relative abundance of given cell types, for example by differential recruitment of immune cell subsets to the site of infection. Accordingly, flow cytometry was used to quantify immune cell subsets present in lung tissue 6 h after infection with either 9–47-Ear, 4559-Blood, 9–47M or 4559M. The surface marker staining panel used (Supplementary Table 4) and appropriate gating strategy (Supplementary Fig. 1) allowed the identification and enumeration of natural killer (NK) cells, neutrophils, eosinophils, inflammatory monocytes (iMono), resident monocytes (rMono), alveolar macrophages (AMΦ), interstitial macrophages (iMΦ), CD11b-negative dendritic cells (CD11b-DC), CD11b-positive dendritic cells (CD11b + DC), T cells and B cells[12]. Of these, neutrophils, by far the most abundant cell type, were present in significantly higher numbers in murine lungs infected with 9–47-Ear (versus 4559-Blood, $p < 0.01$; versus 9–47M, $p < 0.05$) and 4559M (versus 4559-Blood, $p < 0.05$) (Fig. 6), both of which have the D249 *rafR* allele and were cleared from the lungs 24 h post-infection. NK cells were also found to be significantly higher in lungs infected with 9–47-Ear (versus 4559-Blood, $p < 0.01$; versus 9–47M, $p < 0.05$), while eosinophils were raised in 4559M infected lungs (versus 4559-Blood, $p < 0.05$) (Fig. 6). However, since the numbers of NK cells and eosinophils were much lower than neutrophils, their contribution to total lung mRNA and their impact on tissue tropism may be less pronounced. Moreover, the large difference in neutrophil recruitment to the lungs at 6 h post-infection aligns with the differential IL-17 response highlighted in the dual RNA-seq data. The higher number of neutrophils in murine lungs infected with 9–47-Ear and 4559M is likely to facilitate their clearance from this niche.

**Impact of neutrophil depletion and IL-17A neutralization**. The greater abundance of neutrophils in lungs infected with strains containing the D249 *rafR* allele that are cleared from the lungs by 24 h suggests that the recruitment and presence of neutrophils is crucial for bacterial clearance from the lung. Thus, differential neutrophil recruitment might be the underlying mechanism for the observed RafR-dependent tropism. To test this, we investigated the importance of neutrophils for persistence of pneumococci in the lungs in the IN challenge model. Injection of anti-mouse Ly6G was used to deplete neutrophils in 32 mice, alongside an isotype control group treated with rat IgG2a. Neutrophil depletion was confirmed in the blood prior to challenge using flow cytometry (Supplementary Fig. 2), with a 76.35% decrease in neutrophils in the anti-mouse ly6G treated mice, relative to the isotype control-treated group ($p < 0.0001$) (Supplementary Fig. 3A). Mice were then challenged with $10^8$ CFU of each strain, for both treatment groups. Bacterial loads were quantified in the nasopharynx and lungs 24 h post-challenge. No significant differences in bacterial numbers in the nasopharynx were seen between strains within each treatment group (Fig. 7a). Also, for both treatments, the numbers of bacteria in the lungs infected with 4559-Blood and 9–47M were higher than 9–47-Ear and 4559M (Fig. 7b), which is consistent with our previous findings[7]. However, the anti-Ly6G-treated groups showed significantly higher lung bacterial loads compared to their respective isotype controls: 4559-Blood anti-Ly6G versus 4559-Blood

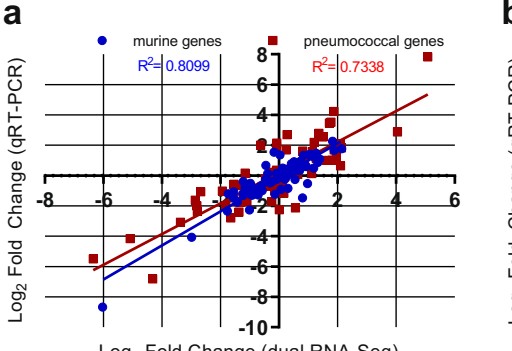
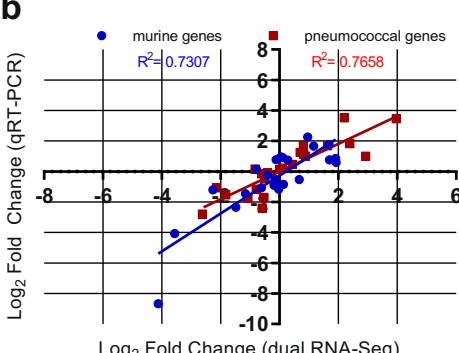

**Fig. 5 Validation of transcriptomic data.** Gene expression values from the dual RNA-seq were confirmed by qRT-PCR, using **a** the same isolated RNA used for the dual RNA-seq or **b** isolated RNA from a repeated experiment. **a** 18 murine and 19 pneumococcal genes were chosen as experimental validation targets. Log₂ fold changes were plotted from qRT-PCR against dual RNA-seq log fold changes for 9–47-Ear versus 4559-Blood, 9–47-Ear versus 9–47M, 9–47M versus 4559M, and 4559-Blood versus 4559M comparisons. A total of 72 murine and 76 pneumococcal comparisons were plotted, with a high degree of correlation observed for both species ($R^2 > 0.73$, Pearson). **b** 6 murine and 6 pneumococcal genes were chosen as targets to test the reproducibility of the dual RNA-seq data. Log₂ fold changes were plotted from qRT-PCR against dual RNA-seq log fold changes for 9–47-Ear versus 4559-Blood, 9–47-Ear versus 9–47M, 9–47M versus 4559M and 4559-Blood versus 4559M comparisons. A total of 24 murine and 24 pneumococcal comparisons were plotted, with a high degree of correlation observed for both species ($R^2 > 0.73$, Pearson).

control ($p < 0.001$), 947-Blood anti-Ly6G versus 947-Blood control ($p < 0.01$), 4559M anti-Ly6G versus 4559M control ($p < 0.01$), and 947M anti-Ly6G versus 947M control ($p < 0.05$) (Fig. 7b). Importantly, the lung bacterial loads of anti-Ly6G-treated 9–47-Ear and 4559M groups were not significantly different to the isotype control-treated 4559-Blood group (Fig. 7b). Thus, restriction of neutrophil infiltration into the lungs by depleting circulating neutrophils in mice challenged with the strains expressing the D249 *rafR* allele resulted in enhanced lung bacterial loads at 24 h similar to that seen in untreated mice challenged with the strains expressing the G249 *rafR* allele.

Given the known involvement of IL-17 in neutrophil recruitment into the lungs after infection[13–16], we also investigated the in vivo significance of the *rafR*-mediated differential expression of IL-17-associated genes between the various *S. pneumoniae* strains. Groups of mice were injected with anti-mouse IL-17A, or a control murine IgG1 antibody, before and after pneumococcal challenge. IL-17A depletion was confirmed by ELISA on separate 9–47-infected bronchoalveolar lavage (BAL) samples, with a 55.63% reduction in IL-17A levels seen in anti-mouse IL-17A treated mice relative to the isotype control group ($p < 0.0001$) (Supplementary Fig. 3b). The effect of this IL-17A depletion on neutrophil levels was also tested using flow cytometry on infected 9–47 lung tissue, with a 22.23% decrease in neutrophils seen in the anti-mouse IL-17A-treated mice, relative to the isotype control-treated group ($p < 0.05$) (Supplementary Fig. 3C). Bacterial loads were quantified in the nasopharynx and lungs for 24 h of post-challenge. Again, no significant differences between strains in bacterial numbers in the nasopharynx were seen within each treatment group (Fig. 7c). However, similar to the results obtained using anti-Ly6G, groups treated with anti-IL-17A showed significantly higher bacterial numbers in the lungs compared to their respective isotype controls; 4559-Blood anti-IL-17A versus 4559-Blood control ($p < 0.05$); 947-Blood anti-IL-17A versus 947-Blood control ($p < 0.05$); 4559M anti-IL-17A versus 4559M control ($p < 0.05$); and 947M anti-IL-17A versus 947M control ($p < 0.01$) (Fig. 7d). Nevertheless, the impact of anti-IL-17 treatment on lung bacterial loads was not quite as dramatic as that of anti-Ly6G, as the number of bacteria in the lungs of anti-IL-17A-treated 9–47-Ear and 4559M groups remained significantly lower relative to the isotype control treated 4559-Blood group (both $p < 0.05$) (Fig. 7d). Together,

these results show that pneumococcal strains carrying the D249 *rafR* allele cause a rapid influx of neutrophils, partly controlled by IL-17 expression in the host, leading to clearance from the lung, while the G249 *rafR* strains manage to remain "stealthy" and hence can persist.

## Discussion

In this study, we have used a dual RNA-seq approach, validated by qRT-PCR, to elucidate the complex interspecies interactions between murine lung cells and infecting *S. pneumoniae* blood and ear isolates that are closely related (same capsular serotype and ST type), but exhibit distinct virulence phenotypes in accordance with their original clinical isolation site. These differences are largely, but not completely, driven by a D249G SNP in the raffinose pathway transcriptional regulator gene *rafR*, which extensively impacts the bacterial transcriptome in the lung environment. The SNP affects expression of genes encoding multiple transmembrane transporters, including those for various sugars, and fine-tunes pneumococcal carbohydrate metabolism. This indicates that the differential expression of sugar catabolism pathways provides specific advantages in distinct host niches, implying differential niche-specific availability of one carbohydrate source versus another. Free sugars are in low abundance in the upper respiratory tract, but *S. pneumoniae* expresses a range of surface-associated exoglycosidases enabling it to scavenge constituent sugars (including galactose, *N*-acetylglucosamine, sialic acid, and mannose) from complex host glycans present in respiratory secretions and on the epithelial surface[17–21]. On the other hand, glucose is readily available in the blood and also in inflamed tissues, implying a marked alteration in the availability of this preferred carbohydrate source as invasive disease progresses[22]. All these variations, and the downstream consequences thereof, are ultimately sensed by host cells, including epithelial and immune cells, resulting in the observed divergence of host response to the various strains, particularly with respect to expression of genes encoding cytokine and chemokine ligands and receptors, as well as those associated with programmed cell death.

Examination of the nature of the host response has provided important clues regarding the mechanism whereby the *rafR* SNP impacts virulence phenotype. By way of example, the dual RNA-seq data showed that expression of IL-17-related genes was

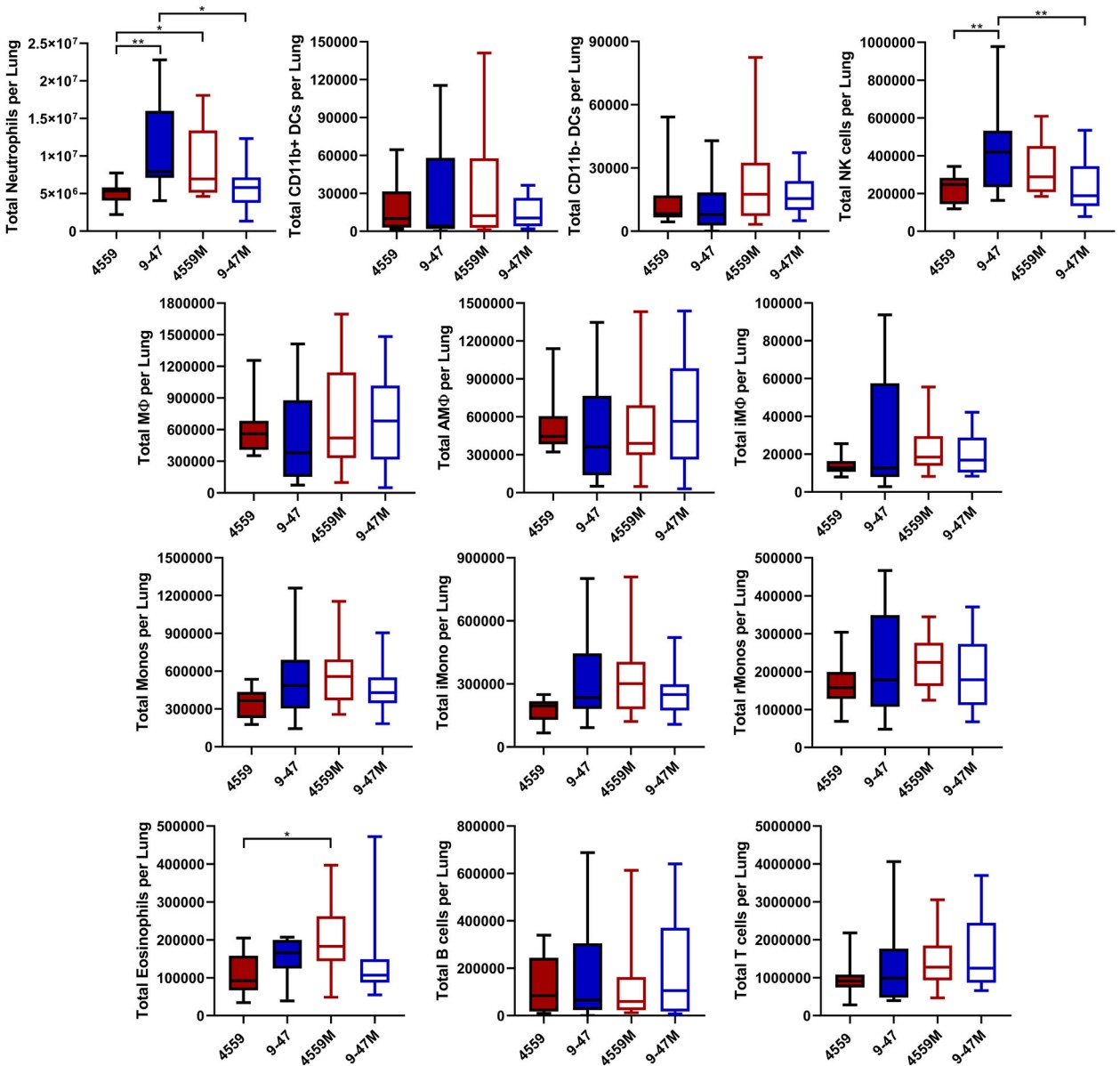

**Fig. 6 Quantification of immune cell subsets in murine lungs 6 h post infection.** Groups of 8 mice per strain were challenged with either 9–47-Ear, 4559-Blood, 9–47M or 4559M. Single cell lung suspensions were prepared and stained with antibodies against various surface markers (Supplementary Table 4) then analyzed by flow cytometry. Populations enumerated include: NK natural killer cells, iMono neutrophils, eosinophils, inflammatory monocytes, rMono resident monocytes, AMΦ alveolar macrophages, iMΦ interstitial macrophages, CD11b−DC CD11b-negative dendritic cells, CD11b+DC CD11b-positive dendritic cells, T cells and B cells. Graphs shown represent pooled data from two independent experiments. All quantitative data are presented as mean ± S.E.M ($n = 16$ for each group), analyzed by one-way ANOVA (*$p < 0.05$; **$p < 0.01$).

enriched in mice infected with 9–47-Ear and 4559M, the strains that express the D249 *rafR* allele and which are cleared from the lungs after 24 h of post-challenge. It is well known that IL-17 drives neutrophil recruitment into the lungs after infection[13–16]. Additionally, neutrophil extravasation genes were shown to be upregulated in murine lungs for 48 h of post pneumococcal challenge[23]. Indeed, we have shown here that neutrophils were present in the lungs at 6 h of post challenge at higher numbers in mice infected with 9–47-Ear and 4559M compared with the strains expressing the G249 *rafR* allele (Fig. 6), as predicted by the dual RNA-seq data. Moreover, we went on to show that neutrophil depletion by treatment with anti-Ly6G increased bacterial numbers in the lungs of mice, relative to the isotype controls. Strikingly, pneumococcal numbers in the lungs of anti-Ly6G

treated mice infected with 9–47-Ear and 4559M were not significantly different to that for isotype control-treated 4559-Blood-infected mice (Fig. 7b). In vivo neutralization of IL-17A also resulted in an increase in bacterial loads in the lungs of mice, relative to the isotype controls, for all challenge strains (Fig. 7d), although not to the same extent as seen with for neutrophil-depleted mice (Fig. 7b). The difference between the impact of IL-17A neutralization versus neutrophil depletion is likely due to the action of alternative neutrophil recruitment pathways[24,25]. Our findings demonstrate that the *rafR* SNP examined in this study has a wide spread effect on both the bacterial and host transcriptomes, with the strains expressing the G249 allele triggering a strong pro-inflammatory IL-17 response in the lungs post-infection. This response leads to an influx of neutrophils to the

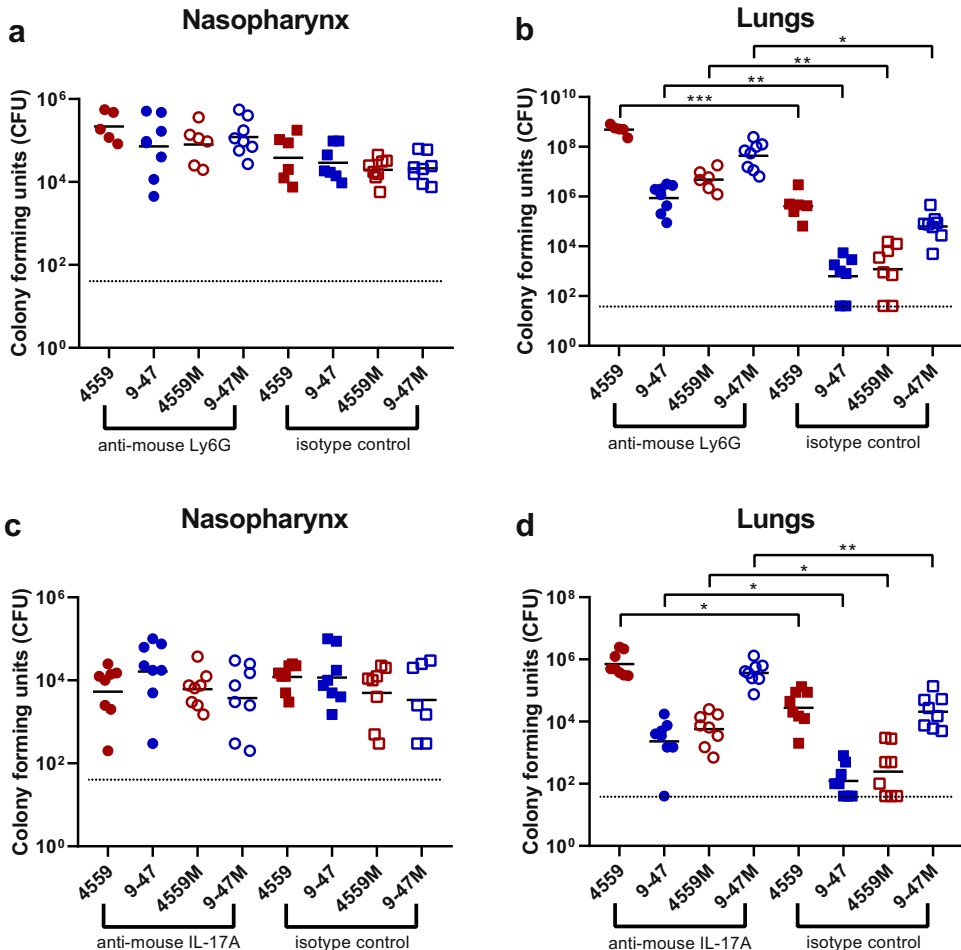

**Fig. 7 Impact of neutrophil depletion or IL-17A neutralization on pneumococcal virulence.** Groups of eight mice were treated with either 350 µg of rat anti-mouse Ly6G or rat IgG2a isotype control, one and two days prior to pneumococcal challenge (**a**, **b**), or with anti-IL-17A or mouse IgG1 isotype control (**c**, **d**), one day before, 2 h before and 6 h after intranasal challenge (see "Methods" section). Twenty-four hours of post-infection, numbers of pneumococci in the nasopharynx and lungs were quantitated (see "Methods" section). NB: $n$ is <8 for some groups because some mice didn't survive the challenge procedure, or until the time of harvest. Viable counts (total CFU per tissue) are shown for each mouse at each site; horizontal bars indicate the geometric mean (GM) CFU for each group; the broken line indicates the threshold for detection. Differences in GM bacterial loads between groups are indicated by asterisks: *$p < 0.05$, **$p < 0.01$, ***$p < 0.001$, by unpaired $t$-test.

lungs, resulting in the clearance of bacteria. Conversely, expression of the D249 *rafR* allele results in a more subdued IL-17 host response, allowing for bacterial persistence in the lungs. Thus, our findings clearly indicate that modulation of neutrophil recruitment during the early stage of infection plays a key role in the capacity of a given *S. pneumoniae* strain to persist in the lungs, and the nature of disease ultimately caused by it.

Our previous studies have shown that in spite of early clearance from the lung, strains 9–47 Ear and 4559M, expressing the G249 *rafR* allele, have an enhanced capacity to spread to and/or proliferate in the ear and brain compartments[6,7]. It is not known whether differential carbohydrate metabolism better adapts these strains to available carbohydrate sources in these niches, or whether altered host pro-inflammatory responses contribute to ascension of the Eustachian tube or penetration of the blood-brain barrier. Unfortunately, the total numbers of pneumococci present in these niches are too low for pathogen–host transcriptomic analyses using available technologies.

Intra-species variation in virulence phenotype is a common feature of pathogenic microorganisms, which by nature are genetically diverse. *S. pneumoniae* is an exemplar of such diversity comprising at least 98 capsular serotypes superimposed on over 12,000 MLST types, and with a core genome that accounts for

only 70% of genes[2]. Nevertheless, stark differences in pathogenic profile can result from the smallest of genetic differences between strains, as exemplified by the profound impact of a single SNP on both bacterial and host transcriptomes reported in this study.

## Methods

**Bacterial strains and growth conditions.** *S. pneumoniae* strains used in this study are listed in Supplementary Table 4. Cells were routinely grown in serum broth (SB) as required. Bacteria were plated on Columbia agar supplemented with 5% (vol/vol) horse blood (BA) and incubated at 37 °C in 5% $CO_2$ overnight.

**Intranasal challenge of mice and extraction of RNA.** Animal experiments were approved by the University of Adelaide Animal Ethics Committee. Groups of 12 outbred 5–6-week-old female Swiss (CD-1) mice (48 in total), were anesthetized by intraperitoneal injection of pentobarbital sodium (Nembutal) and challenged intranasally (IN) with 50 µl of bacterial suspension containing approximately $1 \times 10^8$ CFU in SB of 4559-Blood, 9–47-Ear, 4559M or 9–47M. The challenge dose was confirmed retrospectively by serial dilution and plating on BA. Mice were euthanized by $CO_2$ asphyxiation at 6 h and lungs placed in 1 ml TRIzol (Thermo Fisher). RNA was then extracted using acid-phenol-chloroform-isoamyl alcohol (125:21:1; pH 4.5; Ambion) and purified using the RNeasy minikit (Qiagen). For subsequent dual RNA-seq analyses, there were three replicates per strain, with each replicate derived from the lungs of four mice.

**RNA library preparation and sequencing.** RNA quality was checked using chip-based capillary electrophoresis. Samples were then simultaneously depleted of

murine and pneumococcal ribosomal RNAs by dual rRNA-depletion as previously described[26]. Stranded cDNA library preparation was performed according to the prescribed protocol (Illumina, US). Sequencing was performed for twelve samples in one lane of an Illumina NextSeq 500, High Output Flowcell in 85 single end mode. Libraries were demultiplexed and analyzed further. Raw libraries are accessible at https://www.ncbi.nlm.nih.gov/geo/ with the accession number GSE123982.

**Sequence data analysis**. Quality of raw libraries was checked (FastQC v0.11.8, Babraham Bioinformatics, UK)[27]. In order to improve the quality of alignment, we trimmed the reads using the following criteria: (i) removal of adapter sequence, if any, based on TruSeq3-SE library, (ii) removal of low quality leading and trailing nucleotides, (iii) a five-nucleotide sliding window was created for surviving reads, in which the average quality score must be above 20 and (iv) minimum remaining length must be above 50 (Trimmomatic v0.38)[28]. The quality of trimmed reads was confirmed using FastQC[27].

As reference genomes, we created chimeric genomes by concatenating the in-house generated *S. pneumoniae* strain 9–47-Ear or 4559-Blood circular genomes[7] into the genome of *Mus musculus* (ENSEMBL, release 94, downloaded 9 October 2018). The corresponding annotation file was downloaded at the same time. The chimeric mouse/9–47-Ear genome was used as reference to align libraries from lungs infected by strain 9–47-Ear (and its corresponding swap mutant 9–47M) while the mouse/4559-Blood chimeric genome was used to align 4559-Blood and 4559 M libraries. Notably, the genome of *S. pneumoniae* isolate 4559-Blood has a plasmid. Alignment was performed by RNA-STAR (v2.6.0a)[29] with the following options: (i) alignIntronMax 1 and (ii) sjdbOverhang 84. The aligned reads were then summarized (featureCount v1.6.3) according to the chimeric annotation file in stranded, multimapping (-M), fractionized (--fraction) and overlapping (-O) modes[30]. In order to compare gene expression between strains from ear and blood isolate backgrounds, we prepared a common pneumococcal annotation file using Mauve v20150226[31]. Common genes between 9–47-Ear and 4559-Blood were defined as having common coverage at least 90% and identity at least 90%. This single-pass alignment was selected to minimize false discovery rate. However, due to this approach, we had to adjust the summarizing process, taking into account the overlapping nature of bacterial genes and its organization into operon structures.

We then analyzed host and pathogen libraries separately in R (R v3.5.2). Since reads coming from pneumococcal genes encoding bacterial rRNA dominate the pathogen libraries (average 64.5%; range 61.4–67.3%), we excluded these pneumococcal ribosomal RNA reads from downstream analysis, but we did not do the same exclusion to reads from murine ribosomal RNA genes due to effective rRNA depletion. Differential gene analysis was performed by DESeq2 v1.22.1[32] and genome-wide fold change was calculated within host and pathogen libraries for every two possible comparisons: strains 9–47-Ear to 9–47M, strains 9–47-Ear to 4559-Blood, strains 9–47-Ear to 4559M, strains 9–47M to 4559-Blood, strains 9–47M to 4559M and strains 4559-Blood to 4559M. Value of fold change was set to zero if the corresponding adjusted *p*-value (*p*adj) is reported to be NA.

**Quantitative real time RT-PCR**. Differences in levels of gene expression observed in the dual RNA-seq data were validated by one-step relative quantitative real-time RT-PCR (qRT-PCR) in a Roche LC480 real-time cycler essentially as previously described[33]. The same RNA that was used for the dual RNA-seq was used in the qRT-PCR experimental validation. 19 pneumococcal genes and 18 murine genes were chosen for the experimental validation. The murine intranasal challenge and RNA isolation was repeated to test the reproducibility of the dual RNA-seq data. 6 pneumococcal and 6 murine genes were chosen for the reproducibility validation. The specific primers used for the various genes are listed in Supplementary Table 3 and were used at a final concentration of 200 nM per reaction. As an internal control, primers specific for *gyrA* for pneumococcal genes, and GAPDH for murine genes, were employed. Amplification data were analysed using the comparative critical threshold ($2^{-\Delta\Delta C_T}$) method[34].

**Flow cytometry analysis of infected murine lungs**. Groups of 8 outbred 6-week-old female Swiss (CD-1) mice (32 in total) were anesthetized and challenged with the bacterial suspension as outlined above in the total RNA extraction method. Mice were euthanized by $CO_2$ asphyxiation at 6 h, then lungs were finely macerated in 1 mL prewarmed digestion medium (DMEM + 5% FCS, 10 mM HEPES, 2.5 mM $CaCl_2$, 0.2 U mL$^{-1}$ penicillin/gentamicin, 1 mg mL$^{-1}$ collagenase IA, 30 U mL$^{-1}$ DNase) and incubated at 37 °C for 1 h with mixing every 20 min. Single cells were then prepared for acquisition on a BD LSRFortessa X20 flow cytometer as previously described[35]. The single cell suspensions were stained with antibodies against surface markers listed in Supplementary Table 4, allowing the enumeration of a number of immune cell subsets, as previously described[12].

**Neutrophil depletion, IL-17A blockade and bacterial loads**. Groups of 8 outbred 6-week-old female Swiss (CD-1) mice (64 in total) were intraperitoneally administered with either 350 μg of rat anti-mouse Ly6G or rat IgG2a isotype control antibodies, one and two days prior to pneumococcal challenge, or 200 μg of either monoclonal anti-mouse IL-17A or mouse IgG1 isotype control antibodies one day

prior to, 2 h before and 6 h after pneumococcal challenge. Mice were also cheek bled on the day of challenge for confirmation of depletion of Ly6G-positive cells via flow cytometry, as previously described[36]. Mice were then anesthetized and challenged with the bacterial suspension as outlined above in the total RNA extraction method, for each treatment group. Mice were euthanized by $CO_2$ asphyxiation at 24 h, then nasopharynx and lung tissue samples were harvested and pneumococci enumerated in tissue homogenates as described previously via serial dilution and plating on BA containing gentamicin[37].

**Neutrophil and IL-17A levels after IL-17A neutralization**. Groups of three outbred 6-week-old female Swiss (CD-1) mice were intraperitoneally administered with 200 μg of either monoclonal anti-mouse IL-17A or mouse IgG1 isotype control antibodies one day prior and 2 h before intranasal challenge with 9–47, as described above. Six hours of post-infection, lung tissue and bronchoalveolar lavage (BAL) were harvested. To quantify IL-17 levels, a mouse IL-17 DuoSet® ELISA (R&D Systems) was performed on BAL samples in duplicate, according to the manufacturer's instructions. Flow cytometry was performed on lung tissue samples, as described above, to quantify neutrophil levels after anti-mouse IL-17A treatment.

**Statistics and reproducibility**. For the RNA-seq data, enrichment tests to assess enrichment were performed by the built-in function, *fisher.test()*. Corresponding *p*-values of the enrichment test were adjusted by Bonferroni correction. Resultant figures encompass data derived from three replicates per group, with each replicate derived from lungs of four mice. All other data are presented as mean ± standard error of mean (SEM) or geometric mean, and were analyzed by two-tailed unpaired Student's *t*-test, one way ANOVA or Pearson correlation coefficient, using Prism v8.0d (GraphPad). Statistical significance was defined as *p* < 0.05. Data presented in figures are representative of at least two independent in vivo experiments, or at least 3 independent in vitro experiments.

**Reporting summary**. Further information on research design is available in the Nature Research Reporting Summary linked to this article.

## Data availability

Raw RNA libraries are accessible at https://www.ncbi.nlm.nih.gov/geo/ with the accession number GSE123982. Source data for Figs. 2–7 are provided in Supplementary Data 13.

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

## Acknowledgements

We thank V Benes (GeneCore, EMBL, Heidelberg) for support in library preparation and sequencing and the Center for Information Technology of the University of Groningen for support and for providing access to the Peregrine high-performance computing cluster. We also thank Timona Tyllis and Todd Norton for assistance in acquiring the flow cytometry samples, and Alexandra Tikhomirova for assistance with the murine experiments. This work was supported by the Swiss National Science Foundation (SNSF) (project grant 31003A_172861) to J.W.V., National Health and Medical Research Council (NHMRC) Program Grant 1071659 and Investigator Grant 1174876 to J.C.P. and a University of Adelaide Beacon Fellowship to C.T. The funders had no role in study design, data collection and interpretation, or the decision to submit the work for publication.

## Author contributions

Conceptualization: V.M., J.C.P., J.W.V. and C.T.; Methodology: V.M., R.A., J.W.V. and C.T.; Formal analysis: V.M., R.A., L.J.M., I.C., S.R.M., J.C.P., J.W.V. and C.T.; Investigation: V.M., R.A., H.W., S.C.D., K.T.M., and C.T.; Writing—original draft: V.M., R.A., J.C.P., J.W.V. and C.T.; Writing—review and editing: all authors; Supervision: J.C.P., J.W.V., and C.T.; Funding acquisition: J.C.P., J.W.V., and C.T.

## Competing interests

The authors declare no competing interests.
