## [Peer Review File · Communications Biology]

Reviewers' comments:

Reviewer #1 (Remarks to the Author):

This paper sets out to determine whether a SNP in the *rafR* gene of *S. pneumoniae* influence host-pathogen transcriptional crosstalk. The authors present extensive RNA-seq data that show extensive rewiring of host and pathogen responses related to the *rafR* SNP. The study explores an important area in pathogen biology and has novel results; however, some clarification of the data presented is required.

Major Points

1. Title. The title gives the impression that the RNAseq experiments have totally explained tissue tropism of *S. pneumoniae*, which is not the case. Perhaps better – and highlighting the role of *rafR* – “In vivo RNA-seq reveals extensive *rafR* dependent changes in host and pathogen transcription in a murine model of *S. pneumoniae* infection that contribute to tissue tropism”.
2. Methodology. The number of reads aligned to the bacterial genome is low which is to be expected. I cannot see a reference to an accession number for the strains which were sequenced to provide the alignments – this needs to be clarified or supplied. In addition, although the details of each run are on the GEO data store, it would help readers to be able to see a simple table with the reads per sample and those aligned with mouse, bacteria etc with percentages.
3. The replicates are adequate and the PCA plots show they cluster well, but are there some genes which show big differences between the replicates? Again it would be helpful to see say the top 50 differentially expressed genes between some of the groups as a heat map for each pooled sample to show how consistent the observed differences are.
4. Methods. Why was the fold cut off set at >2 for the pneumococcal genes but > 1.5 for the murine genes? My personal feeling is that this is on the low side and the numbers of genes that are differentially expressed becomes cumbersome to appreciate.
5. Results lines 182-203. These data need to be put into a table as reading them as flowing text is very difficult.
6. Results line 249. No data relating to the non-coding RNAs is provided as far as I can see. Either omit or make clear these data are not contained within the current analysis.
7. Figure 5 and associated text. The authors address the important differences in cell types noted between infection with the different strains. Some additional discussion is required to expand on what differences may thus result in the transcriptional responses observed.
8. Are there data showing changes in IL-17 levels after addition of neutralising antibody? This is not essential if not performed but the text should acknowledge that the effectiveness of the antibody is not entirely clear (although it does clearly have a phenotypic effect).

Reviewer #2 (Remarks to the Author):

This interesting study presents a careful analysis of dual microbe/host RNA seq on an extremely

interesting and potentially informative set of related and in some cases isogenic strains that display differences in tissue tropism. I believe that the results point to important roles for wholesale expression changes in either the pathogen or the host in determining tissue tropism. A current drawback of the current presentation is that it does not provide a clear narrative that permits the reader to understand the important points of the work. As a result, much of the manuscript describes features of the comparative analyses that do not obviously point to new insights but rather give the (incorrect) impression of a laundry list of unrelated comparisons. At the end of this review, I provide a potential alternative framing that the authors should consider. In addition, the authors should address the following points:

1. Fig. 4 is a technical validation that RNA seq reflects the abundance of the RNA species, not an experimental validation that the result is reproducible. To obtain experimental replicates without the expense of RNA seq, the authors should perform separate infections and then utilizing qRT-PCR to validate specific changes in transcription predicted by their RNAseq analyses.
2. The authors should determine how many PMNs are in the lung after anti-IL-17 treatment so as to assess to what degree the phenotype of the treatment can be attributed to PMN infiltration.
3. The authors should clarify whether the only significantly down-regulated genes are the genes listed in lines 189-191. If there are other significantly down-regulated genes by either the bacteria or the host, these genes should be described in the text or in a supplementary file.

To promote an understanding of the rationale of the comparisons performed in the RNA seq analysis, the authors should explicitly present in the Introduction that although a single SNP in the *rafR* gene determines the differential virulence phenotype between 4559 and 9-47, these strains are not isogenic and contain many more non-synonymous SNPs, and that 4559/4559M and 9-47/9-47M are isogenic strains with a single polymorphism in *rafR*. The in the Results, consider the following framework:

1. 4559, the strain that persists in the murine lung, elicits about one-half the number of PMNs at 24 h post-infection than does 9-47, the strain that is cleared from the lung and colonizes the ear and the brain (Fig. 5).
2. Analysis of 4559M and 9-47M suggests that the *rafR* SNP appears partially responsible for this difference.
3. PMN depletion
 - a. Has no effect on NP colonization (as expected?) (Fig. 6)
 - b. Results in higher bacterial loads in the lung for all four strains, confirming multiple previous reports that this host cell is critical for defense against *Sp* lung infection (Fig. 6).
 - c. Permits 9-47 and 4559M to colonize the lung at levels indistinguishable from the pulmonary levels of 4559 in isotype-treated control, suggesting that PMNs are the major factor in the suppressed levels of 9-47 and 4559M in the lung (Fig. 6).
4. Anti-IL-17 treatment has effects similar to (but not as pronounced) as PMN depletion (Fig. 6).
5. RNA seq analysis of bacteria (Fig. 2) shows that
 - a. 9-47 and 4559 show large differences in transcriptional profile, differences that might explain the impact of the *rafR* SNP on tropism in vivo.
 - b. 9-47 and 9-47M show large differences in transcriptional profile and these differences might explain the impact of the SNP on tropism in vivo. (Differences are in CHO metabolism, ABC and sugar transporters.)
 - c. 4559 and 4559M show very few differences in bacterial transcriptional pattern, in spite of the fact that the strains behave differently in vivo, suggesting that the contribution of the *rafR* SNP on tropism is not the result of wholesale bacterial transcription differences. Conversely, 9-47M and 4559 show large differences in transcriptional profile in spite of their similarity in tropism in vivo. Hence, transcriptional profiling is not able to predict tissue tropism in this case.
 - i. Note that the figure legend text should clarify which text refers to panel (b) and which text refers to panel (c).

6. RNA seq analysis of host (Fig. 3) shows that

a. 9-47 and 4559 induce very different host transcriptional profiles, consistent with the different inflammatory responses they elicit (and their different tissue tropism) *in vivo*. In addition, the host transcriptional profile changes dramatically with the *rafR* SNP in both strain backgrounds, consistent with the critical nature of this SNP.

b. Interestingly, the transcriptional response to 4559M closely resembles that of 9-47, suggesting that the *rafR* SNP is largely responsible for the distinct host transcriptional response to 9-47 vs. 4559.

c. In contrast, the transcriptional response to 9-47M does not closely resemble that of 4559, in spite of the observation that the two strains elicit a similar PMN response and respond similarly to PMN depletion or anti-IL-17 treatment. This suggests that, not surprisingly, transcriptional response does not perfectly correlate with inflammatory response or bacterial load.

7. Consistent with the demonstrated differential recruitment of PMN in response to pulmonary challenge with strains displaying different tropism, RNA seq analysis of host (Fig. 3) shows that cytokine/cytokine receptors and IL-17-associated genes were among those differentially regulated.

Minor Concerns

1. Line 348 references the incorrect figure panel.

NB: Reviewers comments are in italics, responses are in plain type.

Reviewer #1 (Remarks to the Author):

This paper sets out to determine whether a SNP in the rafR gene of S. pneumoniae influence host-pathogen transcriptional crosstalk. The authors present extensive RNA-seq data that show extensive rewiring of host and pathogen responses related to the rafR SNP. The study explores an important area in pathogen biology and has novel results; however, some clarification of the data presented is required.

Major Points

1. Title. The title gives the impression that the RNAseq experiments have totally explained tissue tropism of S. pneumoniae, which is not the case. Perhaps better – and highlighting the role of rafR – “In vivo RNA-seq reveals extensive rafR dependent changes in host and pathogen transcription in a murine model of S. pneumoniae infection that contribute to tissue tropism”.

We have modified the title as suggested by the reviewer.

2. Methodology. The number of reads aligned to the bacterial genome is low which is to be expected. I cannot see a reference to an accession number for the strains which were sequenced to provide the alignments – this needs to be clarified or supplied. In addition, although the details of each run are on the GEO data store, it would help readers to be able to see a simple table with the reads per sample and those aligned with mouse, bacteria etc with percentages.

The accession number for RNAseq libraries is specified on line 564; genome sequence accession numbers for 4559 and 9-47 have previously been provided in reference 7 (cited on line 482). Host and pneumococcal read counts and percentages are now shown in new Supplementary Table 1.

3. The replicates are adequate and the PCA plots show they cluster well, but are there some genes which show big differences between the replicates? Again it would be helpful to see say the top 50 differentially expressed genes between some of the groups as a heat map for each pooled sample to show how consistent the observed differences are.

Heat maps displaying the top 50 differentially expressed pneumococcal and murine genes after infection with the various strains (all three replicates) are now shown in new Figure 2. Fold change values of all significantly differentially expressed genes can be found in Supplementary Tables 2-13.

4. Methods. Why was the fold cut off set at >2 for the pneumococcal genes but > 1.5 for the murine genes? My personal feeling is that this is on the low side and the numbers of genes that are differentially expressed becomes cumbersome to appreciate.

The lower FC cut-off for murine genes was chosen to strike a balance between simplicity of analysis and sensitivity (see lines 139-140). A cut-off of 2 for murine genes would have resulted in very few DEGs for some pairwise comparisons (as evident from Supplementary Tables 8-13) which may have reduced the capacity to recognise genes that are commonly-differentially expressed among strains with a given virulence phenotype.

5. Results lines 182-203. These data need to be put into a table as reading them as flowing text is very difficult.

This region of text has now been simplified (see lines 184-191) and the data shown in new Supplementary Table 14.

6. Results line 249. No data relating to the non-coding RNAs is provided as far as I can see. Either omit or make clear these data are not contained within the current analysis.

We have now stated that these data are not presented.

7. Figure 5 and associated text. The authors address the important differences in cell types noted between infection with the different strains. Some additional discussion is required to expand on what differences may thus result in the transcriptional responses observed.

Additional discussion of the differences in cell types and numbers in relation to overall transcriptional responses and clearance from the lungs is now included on lines 323-328.

8. Are there data showing changes in IL-17 levels after addition of neutralising antibody? This is not essential if not performed but the text should acknowledge that the effectiveness of the antibody is not entirely clear (although it does clearly have a phenotypic effect).

IL-17A depletion by the antibody was confirmed by ELISA on separate 9-47-infected bronchoalveolar lavage (BAL) samples, with a 55.63% reduction in IL-17A levels seen in anti-mouse IL-17A treated mice relative to the isotype control group ($p < 0.0001$) (see lines 361-364 and new Supplementary Fig. 3B).

Reviewer #2 (Remarks to the Author):

This interesting study presents a careful analysis of dual microbe/host RNA seq on an extremely interesting and potentially informative set of related and in some cases isogenic strains that display differences in tissue tropism. I believe that the results point to important roles for wholesale expression changes in either the pathogen or the host in determining tissue tropism. A current drawback of the current presentation is that it does not provide a clear narrative that permits the reader to understand the important points of the work. As a result, much of the manuscript describes features of the comparative analyses that do not obviously point to new insights but rather give the (incorrect) impression of a laundry list of unrelated comparisons. At the end of this review, I provide a potential alternative framing that the authors should consider. In addition, the authors should address the following points:

In order to provide a clearer narrative emphasising the take-home messages from this study, we have incorporated a number of additional explanatory statements along the lines of those suggested by Reviewer #2 at multiple points in the text (tracked on the marked copy of the revised manuscript). Having said that, we are reluctant to adopt the suggested batting order for the data presentation, as we wanted to emphasise the strength of dual RNA-seq analysis for identifying potential phenotypically critical transcriptomic changes, which can then be tested experimentally, as we did using PMN and IL-17 depletion in this study. Indeed, the order of data presentation closely follows the order in which the experiments were actually performed. We trust this is acceptable to the Reviewer.

1. *Fig. 4 is a technical validation that RNA seq reflects the abundance of the RNA species, not an experimental validation that the result is reproducible. To obtain experimental replicates without the expense of RNA seq, the authors should perform separate infections and then utilizing qRT-PCR to validate specific changes in transcription predicted by their RNAseq analyses.*

To validate the reproducibility of the transcriptomic data, lung RNA samples 6 h post-infection from a different experiment were analysed with qRT-PCR. Here, 6 pneumococcal and 6 murine genes were chosen, and 24 pneumococcal and 24 murine gene Log₂ FC comparisons were performed. A high degree of correlation was observed, for both pneumococcal ($R^2 > 0.77$, Pearson) and murine genes ($R^2 > 0.73$, Pearson) (see lines 299-304 and new panel B in Fig. 5).

2. *The authors should determine how many PMNs are in the lung after anti-IL-17 treatment so as to assess to what degree the phenotype of the treatment can be attributed to PMN infiltration.*

We have now measured the effect of IL-17A depletion on neutrophil levels using Flow Cytometry on infected 9-47 lung tissue, with a 22.23% decrease in neutrophils seen in the anti-mouse IL-17A treated mice, relative to the isotype control treated group ($p < 0.05$). This is now specified in the text (lines 364-367 and new Supplementary Fig. 3C).

3. *The authors should clarify whether the only significantly down-regulated genes are the genes listed in lines 189-191. If there are other significantly down-regulated genes by either the bacteria or the host, these genes should be described in the text or in a supplementary file.*

These are all the bacterial genes which were differentially expressed in common by strains with the same virulence phenotype (now listed in Supplementary Table 14).

To promote an understanding of the rationale of the comparisons performed in the RNA seq analysis, the authors should explicitly present in the Introduction that although a single SNP in the rafR gene determines the differential virulence phenotype between 4559 and 9-47, these strains are not isogenic and contain many more non-synonymous SNPs, and that 4559/4559M and 9-47/9-47M are isogenic strains with a single polymorphism in rafR.

This clarification is now stated in the Introduction (lines 54-57).

Then in the Results, consider the following framework:

1. *4559, the strain that persists in the murine lung, elicits about one-half the number of PMNs at 24 h post-infection than does 9-47, the strain that is cleared from the lung and colonizes the ear and the brain (Fig. 5).*
2. *Analysis of 4559M and 9-47M suggests that the rafR SNP appears partially responsible for this difference.*
3. *PMN depletion*
 - a. *Has no effect on NP colonization (as expected?) (Fig. 6)*
 - b. *Results in higher bacterial loads in the lung for all four strains, confirming multiple previous reports that this host cell is critical for defense against Sp lung infection (Fig. 6).*

c. Permits 9-47 and 4559M to colonize the lung at levels indistinguishable from the pulmonary levels of 4559 in isotype-treated control, suggesting that PMNs are the major factor in the suppressed levels of 9-47 and 4559M in the lung (Fig. 6).

4. Anti-IL-17 treatment has effects similar to (but not as pronounced) as PMN depletion (Fig. 6).

5. RNA seq analysis of bacteria (Fig. 2) shows that

a. 9-47 and 4559 show large differences in transcriptional profile, differences that might explain the impact of the rafR SNP on tropism in vivo.

b. 9-47 and 9-47M show large differences in transcriptional profile and these differences might explain the impact of the SNP on tropism in vivo. (Differences are in CHO metabolism, ABC and sugar transporters.)

c. 4559 and 4559M show very few differences in bacterial transcriptional pattern, in spite of the fact that the strains behave differently in vivo, suggesting that the contribution of the rafR SNP on tropism is not the result of wholesale bacterial transcription differences. Conversely, 9-47M and 4559 show large differences in transcriptional profile in spite of their similarity in tropism in vivo. Hence, transcriptional profiling is not able to predict tissue tropism in this case.

i. Note that the figure legend text should clarify which text refers to panel (b) and which text refers to panel (c).

6. RNA seq analysis of host (Fig. 3) shows that

a. 9-47 and 4559 induce very different host transcriptional profiles, consistent with the different inflammatory responses they elicit (and their different tissue tropism) in vivo. In addition, the host transcriptional profile changes dramatically with the rafR SNP in both strain backgrounds, consistent with the critical nature of this SNP.

b. Interestingly, the transcriptional response to 4559M closely resembles that of 9-47, suggesting that the rafR SNP is largely responsible for the distinct host transcriptional response to 9-47 vs. 4559.

c. In contrast, the transcriptional response to 9-47M does not closely resemble that of 4559, in spite of the observation that the two strains elicit a similar PMN response and respond similarly to PMN depletion or anti-IL-17 treatment. This suggests that, not surprisingly, transcriptional response does not perfectly correlate with inflammatory response or bacterial load.

7. Consistent with the demonstrated differential recruitment of PMN in response to pulmonary challenge with strains displaying different tropism, RNA seq analysis of host (Fig. 3) shows that cytokine/cytokine receptors and IL-17-associated genes were among those differentially regulated.

See response to general comments above.

Minor Concerns

1. Line 348 references the incorrect figure panel.

This has been corrected.

REVIEWERS' COMMENTS:

Reviewer #2 (Remarks to the Author):

The authors have addressed issues raised in the previous review. This is a very interesting study applying dual RNA seq to a set of related *S. pneumoniae* strains differing in tissue tropism that implicates neutrophil responses differential tissue colonization and disease.